# LLMs as Reverse Engineers? Not Yet on Types and Names

## Abstract

Large Language Models (LLMs) have shown promising potential in reverse engineering tasks such as function name recovery, owing to their ability to generate meaningful identifiers under input conditions. However, existing studies primarily emphasize fine-tuning LLMs for particular applications, often without providing a clear rationale for selecting a given model. To address this gap, we systematically evaluate and quantify the performance of widely used open-source mid-sized LLMs, including CodeLlama, Llama 2, and DeepSeek-R1, on two core reverse engineering tasks: name recovery and type inference. Our experimental results reveal that, without fine-tuning, none of these models achieves a high F1 score in either task. These findings enhance our understanding of the practical utility of LLMs in binary analysis and highlight critical avenues for improving their effectiveness in reverse engineering and related domains.

## 1 Introduction

Large Language Models (LLMs) are proving to be powerful tools for assisting humans, particularly developers in computer science. For example, models such as CodeLlama (Roziere et al., 2023) can understand programming semantics and even perform coding tasks. Beyond software development, LLMs are also being explored in computer security, where they have shown promise in binary reverse engineering tasks such as function summarization (Jin et al., 2023) and symbol name recovery in stripped binaries (Xu et al., 2023; Jiang et al.; Xie et al., 2024). These advances highlight the potential of LLMs to drive progress in program analysis, sparking growing interest among security researchers in leveraging them for reverse engineering.

Despite this progress, evaluating the effectiveness of LLMs in binary reverse engineering remains a challenge. The conventional approach involves fine-tuning existing models, but these models are often selected without a clear rationale (Xie et al., 2024; Jiang et al.). While fine-tuning adapts models to specific tasks and enables performance assessment, the diversity in LLM architectures and training quality leads to widely varying outcomes.

A key issue is the lack of comprehensive benchmarking across existing LLMs. Such evaluations are resource-intensive due to the large number of models and GPU requirements. Moreover, since each model produces responses in different ways, a systematic and standardized evaluation framework is essential for fair comparison.

To address this gap, we evaluate nine prominent LLMs, including Llama2 (Touvron et al., 2023), CodeLlama (Roziere et al., 2023), and Deepseek-R1 (Guo et al., 2025), on reverse engineering tasks. Our experiments show that, initially, all evaluated models achieve very low F1 scores (around 0.03) on name recovery and type inference. However, after fine-tuning, DeepSeek-R1 demonstrates a notable improvement, reaching an F1 score of 0.37. This result underscores the critical role of fine-tuning in enhancing LLM performance. These findings provide valuable insights into the strengths and limitations of current LLMs in binary reverse engineering and lay the foundation for future advancements in this field.

**Contributions.** We make the following contributions.

- **Automatic LLM Experiments:** We develop a fully automated framework to systematically evaluate the reverse engineering performance of nine state-of-the-art LLMs, including CodeLlama and DeepSeek-R1.
- **Extensive Evaluation of Various LLMs:** We perform extensive experiments across diverse input representations, providing a broad and fair comparison of LLM capabilities.
- **Uncovering Characteristics and Limitations of LLMs:** Our study highlights the key advantages and shortcomings of existing LLMs, informing future research directions and guiding the selection of models for fine-tuning in reverse engineering applications.

## 2 BACKGROUND & MOTIVATION

**Problem Definition.** We study the capability of LLMs in name recovery and type inference from stripped binaries.

- **Name recovery** involves predicting human-readable, semantically meaningful names for functions, variables, and arguments to improve program comprehension. While prior works (Jiang et al.; Jin et al., 2022) primarily focused on recovering only the function name from the body, we extend this scope by requiring LLMs to infer all function names appearing inside the function body, which considers every callee function.
- **Type inference** entails identifying and labeling the data types of variables, return values, and function arguments, which is essential for reconstructing the program's semantics.

We follow task formulations from prior research (Banerjee et al., 2021; Pei et al., 2021; He et al., 2018; Xie et al., 2024). The model inputs include assembly code or decompiled code, the two predominant forms of binary representation in reverse engineering.

- We use **decompiled code**, which provides a pseudocode-like structure that exposes stripped variable names and type information more clearly, for both name recovery and type inference.
- We use **assembly code**, as it directly reflects the binary instructions, despite its lack of high-level context and semantic cues, only for the function name recovery of assembly code.

**Binaries.** When a developer writes a program in C, the source code must go through the compilation process to produce an executable binary. During compilation, the binary may include debug symbols, resulting in an *unstripped binaries*. These symbols preserve the original names of variables, functions, and other identifiers, making the binary easier to analyze and debug. However, unstripped binaries are significantly larger in size. To reduce storage overhead and protect intellectual property, developers typically remove these symbols, producing a *stripped binaries*. Stripped binaries retain only the executable instructions but omit the symbolic information, making them smaller and more difficult to reverse engineer. At the same time, the developer can choose the level of optimization to select to what degree the program will be optimized from the source code.

**Related works.** Several studies have explored reverse engineering tasks with AI models, though most focus on narrow aspects. First, Shang et al. (2024) investigated function name recovery and binary code summarization. However, their approach is limited to predicting only the function name from the given code body, without addressing variable name recovery or type inference. Then, Yang et al. (2025) measured AI-augmented decompiler performance on variable naming and type inference. While relevant, their study does not incorporate LLMs and does not address function name recovery. Last, Liu & Wang (2020) examined the performance of traditional decompilers, providing a baseline but not leveraging LLMs for reverse engineering tasks. To our knowledge, no prior work has systematically evaluated LLMs across both function name recovery and type inference, leaving an important gap in understanding their effectiveness.

Moreover, in terms of related reverse engineering works, there are several works conducting function name recovery (Jiang et al.; Jin et al., 2022; David et al., 2020) and variable name recovery (Banerjee et al., 2021; Xie et al., 2024). Furthermore, there are several research papers inferring type inference (Pei et al., 2021; Chen et al., 2022; Zhang et al., 2021; He et al., 2018).

**Motivation.** The rapid growth of open-source LLMs has fueled their adoption in reverse engineering, where they are often fine-tuned for specialized tasks. However, the choice of model is rarely

justified, raising concerns about bias, inconsistency, and reproducibility in research outcomes. For example, SYMGEN (Jiang et al.) employs CodeLlama-34B, while RESYM (Xie et al., 2024) relies on StarCoder-3B, yet the rationale behind these selections is never clearly explained. Such arbitrary choices hinder fair comparisons and make it difficult to understand whether performance differences stem from model design, training data, or task-specific fine-tuning.

In computer science domains, systematic LLM benchmarking has already become common practice, enabling researchers to make informed model selections. By contrast, binary reverse engineering still lacks a rigorous, comparative evaluation framework. As a result, the field risks repeating efforts, overlooking promising models, or drawing misleading conclusions about LLM capabilities.

This gap motivates our work. We conduct the first systematic, large-scale evaluation of multiple state-of-the-art LLMs on two fundamental reverse engineering tasks: name recovery and type inference. By doing so, we aim to provide concrete, evidence-driven insights into model performance, establish fair and reproducible benchmarks, and lay the foundation for principled model selection in future reverse engineering research.

## 3 AUTOMATED MEASUREMENT

Because different LLMs produce responses in varying formats, a standardized evaluation pipeline is required to ensure fair and consistent comparison. To achieve this, we design an automated post-processing workflow that normalizes outputs across models. This pipeline leverages an auxiliary LLM specialized in formatting, which converts heterogeneous responses into a uniform structure suitable for downstream evaluation.

### 3.1 DATA REPRESENTATION

One of our primary input formats is decompiled code as shown in Figure 3, whose structure and symbols are strongly influenced by the decompiler. To eliminate these artifacts, we normalize symbol names by replacing them with placeholders (e.g., VAR, TYPE, and FUNC), each tagged with a unique identifier to distinguish instances. This normalization minimizes noise from decompiler-specific details and allows us to measure LLM capabilities in reverse engineering more directly. The further procedure for generating decompiled code is described in the Appendix.

### 3.2 MODEL INFERENCE

**Response Generation.** We initialize each target pre-trained LLM using the Hugging Face transformers API (tra, 2023). Rather than optimizing prompts, we adopt an established reverse engineering promp (gpt, 2023) that has been shown effective in prior ChatGPT based studies (cha). Figure 4a presents the exact template used to query models. The prompt provides task instructions alongside the input code (decompiled or assembly), explicitly indicating which placeholders require inference.

For example, all placeholders in the decompiled code are included in the prompt, ensuring that the model clearly understands which tokens to predict. Nonetheless, due to differences in instruction following ability, responses often deviate from the requested format, include extraneous explanations, or vary between short tokens and full sentences.

**Response Extraction.** Such inconsistencies make direct evaluation infeasible. Moreover, a single parsing algorithm cannot robustly handle the degree of variation across models. To address this, we employ an auxiliary LLM to post process outputs. The auxiliary model is prompted with the raw response and the formatting specification, producing a normalized result in a consistent schema. This standardization eliminates ambiguities caused by heterogeneous response styles and enables automated, large-scale evaluation of LLM outputs.

### 3.3 LLM FINE-TUNING

To assess LLMs beyond zero-shot inference, we also fine-tune them on our reverse engineering dataset. Full parameter fine-tuning is computationally expensive for large models, so we adopt low-rank adaptation (LoRA) (Hu et al., 2022; Dettmers et al., 2023). LoRA freezes the base model

| LLMs | Size | Years | LLMs | Size | Years |
|---|---|---|---|---|---|
| CodeLlama (Roziere et al., 2023) | 13B | 2023 | Llama2 (Touvron et al., 2023) | 13B | 2023 |
| WizardCoder (Luo et al., 2023) | 15B | 2023 | Deepseek-V2 (Zhu et al., 2024) | 16B | 2024 |
| Qwen 2.5 (Team, 2024) | 14B | 2024 | Phi4 Unsloth (AI, 2024) | 14B | 2024 |
| Qwen 3 (Team, 2025b) | 14B | 2025 | Deepseek-R1-Qwen (Guo et al., 2025) | 14B | 2025 |
| Gemma 3 (Team, 2025a) | 12B | 2025 | | | |

**Table 1: List of Evaluated LLMs.**

weights and introduces trainable low-rank matrices, which are orders of magnitude smaller than the full parameter set Only these matrices are updated during fine-tuning. This approach drastically reduces resource requirements, allowing us to fine-tune multiple models efficiently while preserving the representational capacity of the base LLMs.

### 3.4 EVALUATION METRICS

Evaluating name recovery and type inference requires task-specific metrics to ensure accuracy and fairness.

**Type Inference.** In the C programming language, primitive types (e.g., char, int) must be matched exactly. We therefore apply strict equality checks: the inferred type must match the ground truth completely. This rule extends to user-defined and composite types, where partial matches are not accepted. This strict matching ensures precise measurement of type inference accuracy.

**Name Recovery.** Unlike type inference, evaluating function and variable names is more complex. Developers may use different but semantically equivalent identifiers, including synonyms, abbreviations, or acronyms. To handle this, we adopt the SYMLM (Jin et al., 2022) method, which leverages CodeWordNet to compute semantic distances between sub-names. The method decomposes each function or variable name into sub-words, calculates semantic distances between corresponding sub-words of the original and inferred names, and averages these values. That is, based on the sub-word comparison, the inferred name could obtain partial scores.

## 4 EXPERIMENTS

### 4.1 DATASET

We generate the decompiled code via diverse open-source projects, mostly from GNU software (gnu, 2025). To comprehensively evaluate the capabilities of LLMs, we include binaries compiled for four architectures (x86-64, x86-32, ARM 32-bit, MIPS 32-bit) at multiple compiler optimization levels. Given the large number of available functions and practical time constraints, we randomly select $2,500$ function samples from each architecture and optimization pair, yielding a total of $40,000$ unique functions for evaluation. To guarantee dataset uniqueness, we remove duplicates by comparing both the decompiled code and the corresponding assembly code. Each LLM is then evaluated on this dataset using precision, recall, and F1 score, which are widely accepted metrics for measuring performance in name recovery and type inference tasks.

### 4.2 MODELS

We support eight mainstream code LLMs described in Table 1. It specifies the models, their sizes, and the published years. We select LLMs that are widely deployed for programming tasks (pop, 2023), and well-documented in the literature (Li et al., 2023; Guo et al., 2025). To ensure a fair comparison of existing models, we chose to use models of similar sizes among the various available models. We exclude ChatGPT (cha) because it is not free to utilize its API, and its size is 175 billion for GPT3 and 1.76 trillion for GPT4, which are not adequate for comparison. Similarly, any larger and charged models are excluded from our measurement.

### 4.3 EXPERIMENT SETUP

**Name Recovery and Type Inference via Decompiled Code.** We first evaluate the ability of LLMs to recover both function and variable names, as well as to infer variable types, using the full set of 40,000 samples. Decompiled code serves as input because it retains richer semantic information than assembly code, enabling a more comprehensive assessment of each LLM across all architectures and optimization levels. This approach follows prior work such as VARBERT (Pal et al., 2024) and RESYM (Xie et al., 2024), which also rely on decompiled code for variable name recovery. By using the same setup for both name recovery and type inference, we obtain a deeper understanding of LLM performance in semantic program analysis.

**Function Name Recovery via Assembly Code.** Assembly code provides a faithful representation of the execution process of a function and is therefore suitable for function name recovery. However, as noted earlier, recovering variable names from assembly is significantly more difficult due to the lack of semantic context. Accordingly, this experiment focuses exclusively on predicting function names from assembly, again using the 40,000 sample dataset. Unlike prior work (Jin et al., 2022), we do not incorporate program state information. In summary, the assembly-based setup evaluates only function name recovery.

**Name Recovery With and Without CodeWordNet.** To measure the contribution of CodeWordNet, which plays a central role in our name recovery evaluation, we run comparative experiments with and without CodeWordNet. These experiments are restricted to the x86-64 architecture to illustrate general trends without requiring a full-scale analysis across all architectures. Only the F1 score is reported, as it effectively reflects the trends in both precision and recall observed in earlier experiments.

**Impact of Model Size on Task Performance.** To investigate whether larger models yield better performance, we evaluate three variants of CodeLlama with different parameter sizes. The 70B model is excluded due to GPU limitations. Similar to the CodeWordNet experiments, this analysis is conducted on the x86-64 architecture, with F1 score as the primary metric.

**Performance Evaluation with Fine-Tuned Models.** To assess the benefits of fine-tuning, we prepare an additional training set by randomly selecting 10,000 functions per architecture and optimization pair, independent of the original 2,500 sample test set. This results in a training-testing split of 80% and 20%, a standard practice in machine learning, and a total of 160,000 functions for fine-tuning. Each LLM is fine-tuned on this dataset and evaluated on the held-out test set.

**Evaluation on Real-World Firmware Using Two Decompilers.** Finally, to test LLMs in a realistic reverse engineering setting, we evaluate them on real-world firmware binaries. All symbols are stripped, and the binaries are decompiled into pseudocode using two decompilers: Ghidra and IDA Pro. This allows us to compare performance across decompiler outputs and examine the impact of decompilation quality. Each LLM is evaluated on all tasks (name recovery and type inference) using both versions of the decompiled code.

### 4.4 RESULTS

Due to page limits, we present results primarily on the x86-64 architecture. Results for the remaining architectures are provided in the Appendix.

- **Table 2 and Table 3**: Name recovery performance (original vs. fine-tuned models).
- **Table 4**: Type inference performance (original vs. fine-tuned models).
- **Table 5**: Function name recovery from assembly code.
- **Figure 1**: Impact of CodeWordNet on function name recovery performance.
- **Figure 2**: Comparison of CodeLlama model sizes on name recovery and type inference.
- **Table 6**: Performance on real-world firmware with decompiled code from two different decompilers.

**LLM Performance on Name Recovery.** Overall, original LLMs perform poorly on name recovery tasks, with most models achieving low F1 scores. Function name recovery is generally easier than

| Arch | Model | Function Name Recovery Performance | | | | | | | |
|---|---|---|---|---|---|---|---|---|---|
| | | O0 | | O1 | | O2 | | O3 | |
| | | Orig | Fine | Orig | Fine | Orig | Fine | Orig | Fine |
| | CodeLlama | 0.04 | 0.03 | 0.04 | 0.05 | 0.05 | 0.05 | 0.04 | 0.05 |
| | Llama2 | 0.01 | 0.02 | 0.00 | 0.02 | 0.01 | 0.02 | 0.01 | 0.02 |
| | Deepseek-V2 | 0.01 | 0.03 | 0.02 | 0.04 | 0.02 | 0.04 | 0.02 | 0.04 |
| | Deepseek-R1 | 0.02 | 0.04 | 0.02 | 0.05 | 0.02 | 0.05 | 0.02 | 0.05 |
| x86-64 | Qwen2.5 | 0.03 | 0.04 | 0.03 | 0.05 | 0.03 | 0.06 | 0.03 | 0.06 |
| | Qwen 3 | 0.05 | 0.04 | 0.06 | 0.05 | 0.06 | 0.05 | 0.05 | 0.05 |
| | WizardCoder | 0.01 | 0.04 | 0.01 | 0.06 | 0.01 | 0.06 | 0.01 | 0.06 |
| | Phi4 | 0.04 | 0.05 | 0.05 | 0.07 | 0.05 | 0.07 | 0.04 | 0.07 |
| | Gemma3 | 0.03 | 0.02 | 0.03 | 0.03 | 0.03 | 0.03 | 0.03 | 0.03 |

**Table 2: F1 Scores on Function Name Recovery with x86-64 and Different Optimization Levels.**

| Arch | Model | Variable Name Recovery Performance | | | | | | | |
|---|---|---|---|---|---|---|---|---|---|
| | | O0 | | O1 | | O2 | | O3 | |
| | | Orig | Fine | Orig | Fine | Orig | Fine | Orig | Fine |
| | CodeLlama | 0.01 | 0.22 | 0.02 | 0.08 | 0.01 | 0.05 | 0.01 | 0.08 |
| | Llama2 | 0.01 | 0.26 | 0.01 | 0.05 | 0.00 | 0.04 | 0.01 | 0.06 |
| | Deepseek-V2 | 0.03 | 0.29 | 0.10 | 0.05 | 0.03 | 0.05 | 0.04 | 0.08 |
| | Deepseek-R1 | 0.01 | 0.17 | 0.00 | 0.05 | 0.01 | 0.05 | 0.00 | 0.05 |
| x86-64 | Qwen2.5 | 0.01 | 0.33 | 0.02 | 0.07 | 0.01 | 0.06 | 0.01 | 0.08 |
| | Qwen 3 | 0.02 | 0.18 | 0.04 | 0.06 | 0.02 | 0.05 | 0.02 | 0.05 |
| | WizardCoder | 0.01 | 0.33 | 0.03 | 0.08 | 0.03 | 0.07 | 0.02 | 0.08 |
| | Phi4 | 0.01 | 0.33 | 0.02 | 0.10 | 0.01 | 0.07 | 0.01 | 0.09 |
| | Gemma3 | 0.02 | 0.10 | 0.02 | 0.02 | 0.02 | 0.03 | 0.02 | 0.03 |

**Table 3: F1 Scores on Variable Name Recovery with x86-64 and Different Optimization Levels.**

variable name recovery, though the margin is small. Among original models, Qwen3 achieves the best function name recovery with an F1 score of 0.06. Interestingly, DeepSeek-V2 achieves an F1 score of 0.10 under the O1 optimization setting, but performs worse under other optimization levels.

Fine-tuning significantly improves performance in some cases. For example, WizardCoder improves from 0.01 to 0.33 on variable name recovery after fine-tuning. However, improvements are inconsistent: in some cases (e.g., DeepSeek-V2 under O1), fine-tuning actually reduces performance by half. Still, fine-tuning tends to improve scores across most cases, though the gains are often modest.

> **Finding 1:** Fine-tuning generally improves name recovery, but gains are inconsistent and not guaranteed.

**LLM Performance on Type Inference.** Type inference proves even more challenging, with all models performing worse than on name recovery. Although fine-tuning provides slight improvements, the gains remain marginal. Llama2 consistently outperforms other models, but no model, original or fine-tuned, achieves an F1 score above 0.1. The difficulty can be attributed to strict evaluation criteria requiring exact type matches, as well as the frequent use of user-defined types, which complicates inference further.

| Arch | Model | Variable Type Inference Performance | | | | | | | |
|---|---|---|---|---|---|---|---|---|---|
| | | O0 | | O1 | | O2 | | O3 | |
| | | Orig | Fine | Orig | Fine | Orig | Fine | Orig | Fine |
| | CodeLlama | 0.02 | 0.02 | 0.01 | 0.02 | 0.02 | 0.00 | 0.01 | 0.00 |
| | Llama2 | 0.03 | 0.06 | 0.01 | 0.04 | 0.01 | 0.03 | 0.01 | 0.03 |
| | Deepseek-V2 | 0.01 | 0.00 | 0.01 | 0.00 | 0.01 | 0.00 | 0.00 | 0.00 |
| | Deepseek-R1 | 0.02 | 0.00 | 0.03 | 0.01 | 0.01 | 0.00 | 0.02 | 0.00 |
| x86-64 | Qwen2.5 | 0.00 | 0.01 | 0.01 | 0.00 | 0.00 | 0.00 | 0.01 | 0.00 |
| | Qwen 3 | 0.02 | 0.00 | 0.02 | 0.00 | 0.01 | 0.00 | 0.01 | 0.01 |
| | WizardCoder | 0.00 | 0.00 | 0.00 | 0.00 | 0.00 | 0.00 | 0.00 | 0.00 |
| | Phi4 | 0.02 | 0.00 | 0.01 | 0.00 | 0.01 | 0.01 | 0.01 | 0.00 |
| | Gemma3 | 0.02 | 0.00 | 0.02 | 0.00 | 0.02 | 0.00 | 0.02 | 0.00 |

**Table 4: F1 Scores on Type Inference with x64 and Different Optimization Levels.**

| Model | Function Name Recovery | | Variable Name Recovery | | Type Inference | |
|---|---|---|---|---|---|---|
| | Ghidra | IDA Pro | Ghidra | IDA Pro | Ghidra | IDA Pro |
| CodeLlama | 0.02 | 0.04 | 0.01 | 0.03 | 0.03 | 0.10 |
| Llama2 | 0.00 | 0.01 | 0.01 | 0.01 | 0.02 | 0.12 |
| Deepseek-V2 | 0.00 | 0.00 | 0.03 | 0.01 | 0.01 | 0.02 |
| Deepseek-R1 | 0.03 | 0.03 | 0.01 | 0.02 | 0.07 | 0.05 |
| Qwen2.5 | 0.02 | 0.04 | 0.01 | 0.02 | 0.04 | 0.09 |
| Qwen3 | 0.05 | 0.07 | 0.01 | 0.04 | 0.06 | 0.11 |
| WizardCoder | 0.00 | 0.01 | 0.02 | 0.07 | 0.02 | 0.08 |
| Phi4 | 0.04 | 0.07 | 0.02 | 0.04 | 0.03 | 0.09 |
| Gemma3 | 0.03 | 0.05 | 0.01 | 0.03 | 0.08 | 0.08 |

Table 6: F1 Score on Every Task from Real World Firmware with Two Distinct Decompilers.

> **Finding 2:** Fine-tuning has limited impact on type inference. Improving performance may require richer contextual information and more flexible evaluation metrics.

**LLM Performance on Function Name Recovery via Assembly Code.** Recovering function names from assembly code remains extremely challenging. As shown in Table 5, both original and fine-tuned models achieve 0.0 F1 scores (rounded at the third decimal place). Although models occasionally generate correct predictions, these are too sparse to register improvements. Fine-tuning also fails to produce meaningful gains.

> **Finding 3:** Both original and fine-tuned LLMs struggle with function name recovery from assembly code.

**LLM Performance on Real-World Firmware with Two Decompilers.** On real-world firmware, all models again achieve near-zero F1 scores (Table 6). This outcome is consistent with the weak performance observed in the previous experiments. Interestingly, most models perform slightly better on IDA Pro outputs compared to Ghidra, suggesting that IDA Pro may produce higher-quality or more consistent pseudocode.

**Impact of CodeWordNet.** As shown in Figure 1, removing CodeWordNet causes a sharp drop in performance due to strict string matching penalties. This demonstrates that while LLMs often generate semantically similar names, CodeWordNet is essential for fair evaluation.

| Arch | Model | Asm | |
|---|---|---|---|
| | | O0-3 | |
| | | Orig | Fine |
| x86-64 | CodeLlama | 0.00 | 0.00 |
| | Llama2 | 0.00 | 0.00 |
| | Deepseek-V2 | 0.00 | 0.00 |
| | Deepseek-R1 | 0.00 | 0.00 |
| | Qwen2.5 | 0.00 | 0.00 |
| | Qwen 3 | 0.00 | 0.00 |
| | WizardCoder | 0.00 | 0.00 |
| | Phi4 | 0.00 | 0.00 |
| | Gemma3 | 0.00 | 0.00 |

Table 5: F1 Scores on Function Name Recovery with x86-64 and Different Optimization Levels via Assembly Code.

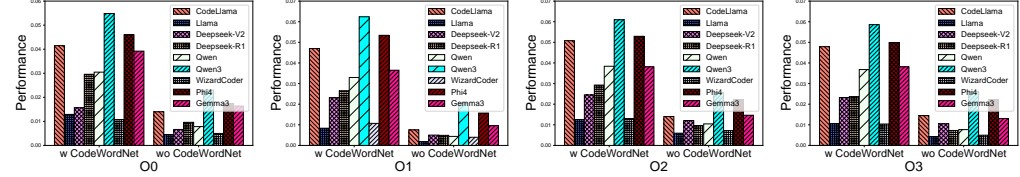

Figure 1: F1 Score on Function Name Recovery with and without CodeWordNet in x86-64 Architecture via Decompiled Code.

**Impact of LLM Model Size vs. Training Quality.** Our observations show that training quality is more important than parameter count. For example, CodeLlama (13B) and Qwen3 (14B) outperform larger models on function name recovery, despite being relatively small.

> **Finding 4:** Larger model size does not guarantee better performance; training data quality is a more decisive factor.

**Impact of LLM Model Architecture vs. Training Quality.** Comparisons of models with identical architectures further highlight the importance of data quality. DeepSeek-R1, a fine-tuned variant of Qwen2.5, consistently underperforms Qwen2.5 in name recovery, despite identical architecture and parameter count. Similarly, CodeLlama outperforms Llama2, likely due to its training optimization for code-related tasks.

> **Finding 5:** Training data quality significantly impacts performance, even among models with the same architecture.

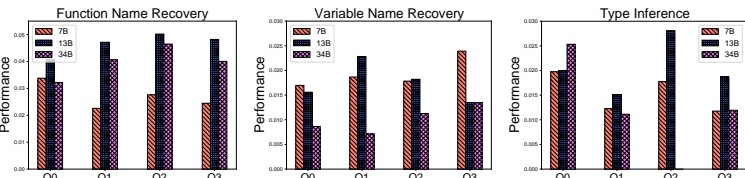

Figure 2: **F1 Score of Name Recovery and Type Inference with Different Sizes of CodeLlama in x86-64 Architecture.**

**Performance of CodeLlama Across Different Model Sizes.** Comparing the 7B, 13B, and 34B variants of CodeLlama (Figure 2) shows little difference in performance. Despite being nearly five times larger, the 34B model does not exhibit significant accuracy gains over the 7B version. While larger models may benefit more from fine-tuning, their raw performance remains largely size-invariant.

> **Finding 6:** Larger variants of the same model family do not necessarily yield better results in reverse engineering tasks.

**Overall LLM Performance.** Across all tasks, baseline LLMs demonstrate limited capability, with most achieving F1 scores below 0.1. While fine-tuning leads to measurable improvements, the gains remain insufficient for LLMs to reliably perform reverse engineering tasks.

### 4.5 FURTHER ANALYSIS

In this section, we provide a deeper analysis of the LLM performance.

| Model | Fine-tuning Time (s/200 steps) |
|---|---|
| CodeLlama | 56,640 |
| Llama2 | 60,540 |
| Deepseek-V2 | 186,720 |
| Deepseek-R1 | 46,740 |
| Qwen2.5 | 48,540 |
| Qwen3 | 74,580 |
| WizardCoder | 63,000 |
| Phi4 | 33,120 |
| Gemma3 | 142,680 |

Table 7: **Time Spent on the Fine-tuning Process**

**Time Spent on Fine-tuning.** As shown in Table 7, most models require approximately 60,000 seconds (16.6 hours) to complete 200 fine-tuning steps. Notably, Qwen2.5 and DeepSeek-R1 finish in around 46,000 seconds, roughly 25% faster than other models. This is expected, since DeepSeek-R1 is built on Qwen2.5. In contrast, DeepSeek-V2 takes 186,720 seconds, over three times longer than the average.

Further investigation revealed that this extended duration is largely due to prompt template design. When we modified CodeLlama's prompt template, we observed a similar increase in training time, confirming the strong influence of template structure. Unfortunately, not all models provide well-documented templates, making it difficult to select effective ones or verify their legitimacy.

> **Finding 7:** Fine-tuning time is highly sensitive to prompt templates; poor template design can increase training duration by up to 4 times longer.

| Original | Finetuned | Ground Truth |
|---|---|---|
| execute tas k | oss l rsa pkey set key | oss l md sha update |
| initial cond ition de termin altion | oss l prov be run | oss l prov be run |
| error hand ling | stack chk fail | stack chk fail |
| main te str out ine | sm t est group | sm sig t est |

**Table 8: Function Name Recovery Cases that the Original and Finetuned Qwen2.5 Model Generate**

**Responses from LLMs.** Despite being instructed to produce outputs in a strict format without explanations, no LLM consistently adhered to the required structure. For example, DeepSeek-R1 frequently generated a chain of thought reasoning before producing an answer. In some cases, the reasoning was so long that the model failed to return the final output. Moreover, it usually causes significant performance overhead due to the length of reasoning. To address this, we modified the prompt template to suppress reasoning, since DeepSeek-R1 lacks a built-in mechanism for disabling it.

Additionally, DeepSeek-R1 occasionally produced generic refusals (e.g., "I'm sorry, but I can't assist with that request."), which prevented task completion. To standardize outputs across all models, we implemented a post-processing step using another LLM to extract only the relevant information. Alternatively, fine-tuning resolved this issue directly, after training, models consistently adhered to the required output format.

---

**Finding 8:** LLMs usually fail to follow consistent output formatting unless assisted by fine-tuning or post-processing.

---

**Cases Where Fine-Tuning Helps.** As discussed in §4.4, fine-tuning improves both performance and formatting. To illustrate this, we highlight cases where the fine-tuned Qwen2.5 model succeeded in function name recovery while the original version failed. Qwen2.5 is particularly notable because it shows substantial improvements after fine-tuning.

Table 8 presents four representative examples. For instance, the original Qwen2.5 model fails to identify the __stack_chk_fail function, which is a common and recognizable function for detecting stack overflow, whereas the fine-tuned version recovers it correctly. Interestingly, the model does not always generate the same prediction for this function, as it frequently appears at the end of decompiled code and often co-occurs with other functions in our dataset. Similarly, functions from the OpenSSL library are more reliably identified by the fine-tuned model, reflecting improved semantic understanding and code recognition.

**Additional Performance Reasoning.** In many cases, LLMs fail to generate the complete name or type requested by the prompt, which inherently lowers F1 scores. For example, although a prompt asks for ten variable names, LLMs often miss some of them, which can lead to a lower F1 score. Moreover, due to their tendency to produce extended reasoning, models sometimes exceed output size limits, causing truncation before the answer is generated. A few models, such as CodeLlama, further complicate evaluation by producing multiple candidate names for a single function, making it unclear which should be considered correct.

## 5  CONCLUSION

Recent years have seen a surge of reverse engineering approaches leveraging machine learning and, increasingly, large language models. While LLMs are often fine-tuned for reverse engineering tasks, the choice of a specific model is typically made without a clear rationale. To address this gap, we conducted a systematic evaluation of nine widely used LLMs on name recovery and type inference. Our experiments reveal that, although fine-tuning can yield measurable improvements, current LLMs remain fundamentally limited in their ability to perform reverse engineering tasks. These findings underscore the need for more effective fine-tuning strategies, higher-quality training data, and possibly alternative input representations to fully unlock the potential of LLMs in binary analysis.

**Reproducibility.** The result can be reproducible by executing the identical scripts. The number will be similar, but not identical, due to randomness of LLMs and random data selection.

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

## A DECOMPILED CODE INPUT GENERATION

Our experiments require two types of input, assembly code and decompiled code. To construct ground truth, we rely on binaries with debug symbols, while the actual input to the models is generated from stripped binaries. This process involves two main steps, collecting variable and function information from unstripped binaries, and then using this information to produce the decompiled code for stripped binaries. Because the assembly code remains identical between stripped and unstripped binaries, we obtain it directly through the decompiler API. The decompiled code, however, requires additional processing.

**Step 1: Extracting Ground Truth from Unstipped Binaries.** For each function in the unstripped binaries, we record the storage location of every variable. Typically, local variables are allocated on the stack according to their size, while function parameters are passed through registers. Using a decompiler API script, we track and store both local variables (with their stack layout) and parameters (with their corresponding registers). The ground truth at this stage consists of variable names, types, and stack and register locations. Additionally, we collect information about all callee functions, including their addresses and names.

**Step 2: Generating Decompiled Code from Stripped Binaries.** Next, we analyze the stripped binaries using the previously constructed ground truth. For each function, we identify stripped variables and match them with their stack addresses to recover their original annotations from the ground truth.

Since the core tasks of our study involve type inference and name recovery, the input code presented to LLMs must be unambiguous. To achieve this, we replace all existing function and variable names in the decompiled code with standardized placeholders. This ensures that the models cannot rely on any residual naming hints and must instead infer types and recover meaningful names from the code context. Figure 3 illustrates the outcome of this process.

```
undefined FUNC1(void)
{
  long in_FS_OFFSET;
  TYPE3 VAR3;
  TYPE2 VAR2;
  TYPE1 VAR1;
  VAR1 = *(long *)(in_FS_OFFSET + 0x28);
  FUNC3(4,0);
  FUNC5(2,FUNC4,&VAR3);
  if (VAR1 == *(long *)(in_FS_OFFSET + 0x28)) {
    return 0;
  }
  FUNC2();
}
```

**Figure 3: Decompiled Code Example of Input Format**

## B PROMPTS USED FOR AUTOMATED MEASUREMENTS

There are two prompts used for automated measurements, 1) Asking LLMs for inference, and 2) Asking a LLM for unification.

## C LLM INPUT TEMPLATES

Since each LLM takes an input following its own template, we need to define it differently. We select the following templates for each LLM.

- **CdeLlama**: `[INST]\n<<SYS>>{instruction}\n<</SYS>>\n\n{input}[/INST]`
- **Llama2**: `[INST]\n<<SYS>>{instruction}\n<</SYS>>\n\n{input}[/INST]`
- **Deepseek-V2**: `<|begin_of_sentence|>User: {instruction}\n{input}\n\nAssistant:`

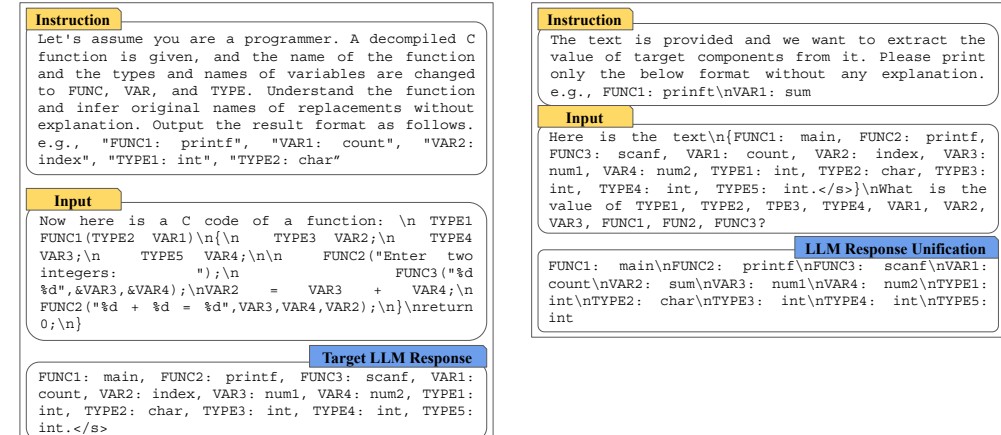

**(a) The Prompt Used to Infer Types/Names**  **(b) The prompt Used to Extract Responses**

**Figure 4: Two Prompts Used for LLM Measurement**

- **Deepseek-R1**: `<|begin_of_sentence|><|User|>{instruction}\n{input}<|Assistant|><think>\n</think>\n\n`
- **Qwen2.5**: `<|im_start|>system\nYou are a helpful assistant.<|im_end|>\n <|im_start|>user\n{instruction}\n{input}<|im_end|>\n<|im_start|>assistant\n`
- **Qwen3**: `<|im_start|>user\nYou are a helpful assistant.\n{instruction}\n{input} <|im_end|>\n<|im_start|>assistant<think>\n\n</think>\n\n`
- **WizardCoder**: `Below is an instruction that describes a task. Write a response that appropriately completes the request.\n\n### Instruction: {instruction}\n\n{input}\n\n### Response:\n`
- **Phi4**: `<|im_start|>user<|im_sep|> You are a helpful assistant.\n{instruction}\n{input}<|im_end|>\n<|im_start|>assistant<|im_sep|>`
- **Gemma3**: `<start_of_turn>user\nYou are a helpful assistant.\n{instruction}\n{input}<end_of_turn>\n<start_of_turn>model\n`

## D  F1 Scores on the Rest Architectures

This section contains F1 scores of every evaluated LLM on the rest architectures, x86, ARM 32-bit, and MIPS 32-bit, as shown in Table 9, Table 10, Table 11, and Table 12, which present the similar trends illustrated in the main context.

## E  Impact of CodeWordNet

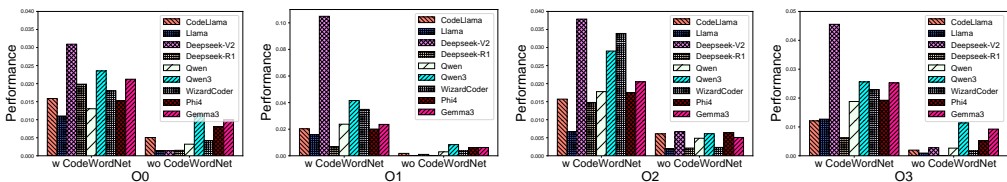

**Figure 5: F1 Score on Variable Name Recovery with and without CodeWordNet in x86-64 Architecture via Decompiled Code.**

| Arch | Model | Function Name Recovery Performance | | | | | | | |
|---|---|---|---|---|---|---|---|---|---|
| | | O0 | | O1 | | O2 | | O3 | |
| | | Orig | Fine | Orig | Fine | Orig | Fine | Orig | Fine |
| x86-32 | CodeLlama | 0.05 | 0.04 | 0.06 | 0.05 | 0.06 | 0.05 | 0.06 | 0.05 |
| | Llama2 | 0.01 | 0.03 | 0.01 | 0.03 | 0.01 | 0.03 | 0.01 | 0.03 |
| | Deepseek-V2 | 0.02 | 0.03 | 0.02 | 0.04 | 0.03 | 0.05 | 0.03 | 0.04 |
| | Deepseek-R1 | 0.02 | 0.06 | 0.03 | 0.07 | 0.02 | 0.07 | 0.03 | 0.07 |
| | Qwen2.5 | 0.03 | 0.05 | 0.04 | 0.06 | 0.04 | 0.06 | 0.04 | 0.06 |
| | Qwen 3 | 0.06 | 0.06 | 0.07 | 0.06 | 0.07 | 0.06 | 0.07 | 0.06 |
| | WizardCoder | 0.01 | 0.05 | 0.02 | 0.06 | 0.02 | 0.06 | 0.02 | 0.06 |
| | Phi4 | 0.04 | 0.05 | 0.06 | 0.07 | 0.06 | 0.07 | 0.06 | 0.07 |
| | Gemma3 | 0.03 | 0.02 | 0.04 | 0.03 | 0.04 | 0.03 | 0.04 | 0.03 |
| ARM | CodeLlama | 0.04 | 0.01 | 0.05 | 0.02 | 0.04 | 0.02 | 0.05 | 0.02 |
| | Llama2 | 0.01 | 0.01 | 0.01 | 0.02 | 0.01 | 0.02 | 0.02 | 0.02 |
| | Deepseek-V2 | 0.01 | 0.03 | 0.02 | 0.03 | 0.02 | 0.03 | 0.02 | 0.04 |
| | Deepseek-R1 | 0.02 | 0.04 | 0.03 | 0.04 | 0.03 | 0.05 | 0.04 | 0.05 |
| | Qwen2.5 | 0.03 | 0.04 | 0.03 | 0.04 | 0.04 | 0.05 | 0.04 | 0.05 |
| | Qwen 3 | 0.06 | 0.03 | 0.07 | 0.04 | 0.07 | 0.04 | 0.07 | 0.04 |
| | WizardCoder | 0.01 | 0.03 | 0.02 | 0.03 | 0.01 | 0.03 | 0.02 | 0.03 |
| | Phi4 | 0.04 | 0.05 | 0.04 | 0.05 | 0.05 | 0.06 | 0.05 | 0.06 |
| | Gemma3 | 0.04 | 0.01 | 0.04 | 0.02 | 0.05 | 0.02 | 0.05 | 0.02 |
| MIPS | CodeLlama | 0.02 | 0.00 | 0.02 | 0.01 | 0.02 | 0.01 | 0.02 | 0.01 |
| | Llama2 | 0.01 | 0.00 | 0.01 | 0.01 | 0.01 | 0.01 | 0.01 | 0.01 |
| | Deepseek-V2 | 0.01 | 0.01 | 0.00 | 0.01 | 0.00 | 0.01 | 0.00 | 0.01 |
| | Deepseek-R1 | 0.02 | 0.02 | 0.02 | 0.02 | 0.02 | 0.02 | 0.02 | 0.02 |
| | Qwen2.5 | 0.02 | 0.01 | 0.02 | 0.02 | 0.02 | 0.01 | 0.02 | 0.02 |
| | Qwen 3 | 0.04 | 0.01 | 0.03 | 0.01 | 0.02 | 0.01 | 0.03 | 0.01 |
| | WizardCoder | 0.01 | 0.02 | 0.00 | 0.03 | 0.00 | 0.03 | 0.00 | 0.03 |
| | Phi4 | 0.03 | 0.02 | 0.02 | 0.02 | 0.02 | 0.02 | 0.03 | 0.03 |
| | Gemma3 | 0.03 | 0.01 | 0.03 | 0.01 | 0.02 | 0.01 | 0.03 | 0.01 |

Table 9: **Performance Evaluation on Function Name Recovery with Different Architectures and Optimization Levels.**

| Arch | Model | Variable Name Recovery Performance | | | | | | | |
|---|---|---|---|---|---|---|---|---|---|
| | | O0 | | O1 | | O2 | | O3 | |
| | | Orig | Fine | Orig | Fine | Orig | Fine | Orig | Fine |
| x86-32 | CodeLlama | 0.03 | 0.04 | 0.04 | 0.06 | 0.04 | 0.06 | 0.03 | 0.06 |
| | Llama2 | 0.02 | 0.02 | 0.02 | 0.02 | 0.02 | 0.01 | 0.02 | 0.01 |
| | Deepseek-V2 | 0.04 | 0.05 | 0.05 | 0.08 | 0.04 | 0.06 | 0.05 | 0.07 |
| | Deepseek-R1 | 0.02 | 0.09 | 0.03 | 0.09 | 0.03 | 0.08 | 0.05 | 0.08 |
| | Qwen2.5 | 0.03 | 0.08 | 0.04 | 0.10 | 0.04 | 0.09 | 0.04 | 0.10 |
| | Qwen 3 | 0.05 | 0.08 | 0.06 | 0.10 | 0.06 | 0.09 | 0.07 | 0.09 |
| | WizardCoder | 0.03 | 0.07 | 0.04 | 0.10 | 0.03 | 0.09 | 0.03 | 0.09 |
| | Phi4 | 0.05 | 0.11 | 0.06 | 0.11 | 0.07 | 0.10 | 0.07 | 0.11 |
| | Gemma3 | 0.03 | 0.03 | 0.04 | 0.05 | 0.04 | 0.03 | 0.04 | 0.04 |
| ARM | CodeLlama | 0.01 | 0.12 | 0.03 | 0.06 | 0.02 | 0.06 | 0.02 | 0.07 |
| | Llama2 | 0.01 | 0.20 | 0.01 | 0.07 | 0.00 | 0.09 | 0.01 | 0.11 |
| | Deepseek-V2 | 0.01 | 0.23 | 0.04 | 0.09 | 0.03 | 0.11 | 0.05 | 0.13 |
| | Deepseek-R1 | 0.00 | 0.18 | 0.03 | 0.08 | 0.01 | 0.09 | 0.01 | 0.11 |
| | Qwen2.5 | 0.01 | 0.28 | 0.02 | 0.10 | 0.01 | 0.13 | 0.02 | 0.15 |
| | Qwen 3 | 0.02 | 0.19 | 0.03 | 0.08 | 0.03 | 0.10 | 0.02 | 0.10 |
| | WizardCoder | 0.02 | 0.27 | 0.04 | 0.10 | 0.02 | 0.12 | 0.03 | 0.17 |
| | Phi4 | 0.02 | 0.28 | 0.03 | 0.10 | 0.03 | 0.12 | 0.02 | 0.14 |
| | Gemma3 | 0.02 | 0.09 | 0.03 | 0.04 | 0.03 | 0.04 | 0.03 | 0.05 |
| MIPS | CodeLlama | 0.01 | 0.01 | 0.02 | 0.01 | 0.03 | 0.01 | 0.02 | 0.01 |
| | Llama2 | 0.01 | 0.01 | 0.02 | 0.00 | 0.02 | 0.00 | 0.02 | 0.01 |
| | Deepseek-V2 | 0.01 | 0.01 | 0.01 | 0.01 | 0.02 | 0.01 | 0.01 | 0.00 |
| | Deepseek-R1 | 0.00 | 0.03 | 0.00 | 0.01 | 0.00 | 0.01 | 0.00 | 0.01 |
| | Qwen2.5 | 0.01 | 0.02 | 0.01 | 0.01 | 0.00 | 0.01 | 0.00 | 0.01 |
| | Qwen 3 | 0.02 | 0.02 | 0.01 | 0.01 | 0.02 | 0.01 | 0.02 | 0.01 |
| | WizardCoder | 0.01 | 0.02 | 0.03 | 0.02 | 0.03 | 0.02 | 0.02 | 0.02 |
| | Phi4 | 0.01 | 0.04 | 0.01 | 0.03 | 0.01 | 0.03 | 0.01 | 0.03 |
| | Gemma3 | 0.02 | 0.01 | 0.02 | 0.01 | 0.03 | 0.01 | 0.03 | 0.01 |

Table 10: **Performance Evaluation on Variable Name Recovery with Different Architectures and Optimization Levels.**

Figure 5 describes the impact of CodeWordNet for variable name recovery. It follows a similar trend to function name recovery.

| Arch | Model | Variable Type Inference Performance | | | | | | | |
|---|---|---|---|---|---|---|---|---|---|
| | | O0 | | O1 | | O2 | | O3 | |
| | | Orig | Fine | Orig | Fine | Orig | Fine | Orig | Fine |
| x86-32 | CodeLlama | 0.02 | 0.03 | 0.02 | 0.03 | 0.3 | 0.02 | 0.02 | 0.01 |
| | Llama2 | 0.03 | 0.05 | 0.01 | 0.06 | 0.03 | 0.05 | 0.01 | 0.03 |
| | Deepseek-V2 | 0.00 | 0.00 | 0.02 | 0.00 | 0.02 | 0.00 | 0.02 | 0.00 |
| | Deepseek-R1 | 0.00 | 0.00 | 0.00 | 0.00 | 0.00 | 0.01 | 0.01 | 0.01 |
| | Qwen2.5 | 0.00 | 0.00 | 0.01 | 0.00 | 0.02 | 0.01 | 0.00 | 0.00 |
| | Qwen 3 | 0.02 | 0.00 | 0.03 | 0.01 | 0.03 | 0.03 | 0.03 | 0.00 |
| | WizardCoder | 0.00 | 0.00 | 0.00 | 0.01 | 0.01 | 0.01 | 0.00 | 0.01 |
| | Phi4 | 0.00 | 0.00 | 0.02 | 0.00 | 0.01 | 0.01 | 0.02 | 0.00 |
| | Gemma3 | 0.01 | 0.00 | 0.01 | 0.00 | 0.04 | 0.00 | 0.03 | 0.00 |
| ARM | CodeLlama | 0.02 | 0.02 | 0.03 | 0.01 | 0.05 | 0.03 | 0.04 | 0.02 |
| | Llama2 | 0.04 | 0.07 | 0.03 | 0.04 | 0.04 | 0.07 | 0.03 | 0.05 |
| | Deepseek-V2 | 0.02 | 0.00 | 0.02 | 0.00 | 0.03 | 0.00 | 0.02 | 0.00 |
| | Deepseek-R1 | 0.03 | 0.00 | 0.01 | 0.00 | 0.08 | 0.00 | 0.03 | 0.01 |
| | Qwen2.5 | 0.01 | 0.02 | 0.01 | 0.00 | 0.03 | 0.01 | 0.01 | 0.01 |
| | Qwen 3 | 0.03 | 0.00 | 0.02 | 0.01 | 0.03 | 0.00 | 0.02 | 0.00 |
| | WizardCoder | 0.00 | 0.01 | 0.01 | 0.00 | 0.00 | 0.01 | 0.01 | 0.01 |
| | Phi4 | 0.02 | 0.02 | 0.04 | 0.00 | 0.04 | 0.00 | 0.02 | 0.01 |
| | Gemma3 | 0.03 | 0.00 | 0.03 | 0.00 | 0.04 | 0.01 | 0.02 | 0.00 |
| MIPS | CodeLlama | 0.03 | 0.03 | 0.02 | 0.02 | 0.01 | 0.02 | 0.02 | 0.02 |
| | Llama2 | 0.04 | 0.06 | 0.03 | 0.05 | 0.02 | 0.07 | 0.03 | 0.06 |
| | Deepseek-V2 | 0.02 | 0.00 | 0.02 | 0.00 | 0.01 | 0.00 | 0.02 | 0.00 |
| | Deepseek-R1 | 0.05 | 0.00 | 0.05 | 0.01 | 0.03 | 0.00 | 0.04 | 0.00 |
| | Qwen2.5 | 0.01 | 0.01 | 0.03 | 0.01 | 0.02 | 0.00 | 0.02 | 0.00 |
| | Qwen 3 | 0.03 | 0.00 | 0.07 | 0.00 | 0.07 | 0.01 | 0.06 | 0.01 |
| | WizardCoder | 0.01 | 0.01 | 0.02 | 0.00 | 0.01 | 0.00 | 0.02 | 0.00 |
| | Phi4 | 0.05 | 0.00 | 0.07 | 0.00 | 0.08 | 0.00 | 0.08 | 0.01 |
| | Gemma3 | 0.03 | 0.00 | 0.03 | 0.01 | 0.04 | 0.00 | 0.03 | 0.00 |

**Table 11: Performance Evaluation on Type Inference with Different Architectures and Optimization Levels.**

| Arch | Model | Function Name Recovery Performance | | | | | | | |
|---|---|---|---|---|---|---|---|---|---|
| | | O0 | | O1 | | O2 | | O3 | |
| | | Orig | Fine | Orig | Fine | Orig | Fine | Orig | Fine |
| x86-32 | CodeLlama | 0.00 | 0.00 | 0.00 | 0.00 | 0.00 | 0.00 | 0.00 | 0.00 |
| | Llama2 | 0.00 | 0.00 | 0.00 | 0.00 | 0.00 | 0.00 | 0.00 | 0.00 |
| | Deepseek-V2 | 0.00 | 0.00 | 0.00 | 0.00 | 0.00 | 0.00 | 0.00 | 0.00 |
| | Deepseek-R1 | 0.00 | 0.00 | 0.00 | 0.00 | 0.00 | 0.00 | 0.00 | 0.00 |
| | Qwen2.5 | 0.00 | 0.00 | 0.00 | 0.00 | 0.00 | 0.00 | 0.00 | 0.00 |
| | Qwen 3 | 0.00 | 0.00 | 0.00 | 0.00 | 0.00 | 0.00 | 0.00 | 0.00 |
| | WizardCoder | 0.00 | 0.00 | 0.00 | 0.00 | 0.00 | 0.00 | 0.00 | 0.00 |
| | Phi4 | 0.00 | 0.00 | 0.00 | 0.00 | 0.00 | 0.00 | 0.00 | 0.00 |
| | Gemma3 | 0.00 | 0.00 | 0.00 | 0.00 | 0.00 | 0.00 | 0.00 | 0.00 |
| ARM | CodeLlama | 0.00 | 0.02 | 0.00 | 0.02 | 0.00 | 0.03 | 0.00 | 0.04 |
| | Llama2 | 0.00 | 0.01 | 0.00 | 0.02 | 0.00 | 0.02 | 0.00 | 0.04 |
| | Deepseek-V2 | 0.00 | 0.00 | 0.00 | 0.00 | 0.00 | 0.00 | 0.00 | 0.00 |
| | Deepseek-R1 | 0.00 | 0.00 | 0.00 | 0.00 | 0.00 | 0.00 | 0.00 | 0.00 |
| | Qwen2.5 | 0.00 | 0.01 | 0.00 | 0.02 | 0.00 | 0.02 | 0.00 | 0.03 |
| | Qwen 3 | 0.00 | 0.02 | 0.00 | 0.03 | 0.00 | 0.03 | 0.00 | 0.04 |
| | WizardCoder | 0.00 | 0.01 | 0.00 | 0.02 | 0.00 | 0.02 | 0.00 | 0.04 |
| | Phi4 | 0.00 | 0.02 | 0.00 | 0.03 | 0.00 | 0.04 | 0.00 | 0.06 |
| | Gemma3 | 0.00 | 0.00 | 0.00 | 0.01 | 0.00 | 0.01 | 0.00 | 0.02 |
| MIPS | CodeLlama | 0.00 | 0.00 | 0.00 | 0.00 | 0.00 | 0.00 | 0.00 | 0.00 |
| | Llama2 | 0.00 | 0.00 | 0.00 | 0.00 | 0.00 | 0.00 | 0.00 | 0.00 |
| | Deepseek-V2 | 0.00 | 0.00 | 0.00 | 0.00 | 0.00 | 0.00 | 0.00 | 0.00 |
| | Deepseek-R1 | 0.00 | 0.00 | 0.00 | 0.00 | 0.00 | 0.00 | 0.00 | 0.00 |
| | Qwen2.5 | 0.00 | 0.00 | 0.00 | 0.00 | 0.00 | 0.00 | 0.00 | 0.00 |
| | Qwen 3 | 0.00 | 0.00 | 0.00 | 0.00 | 0.00 | 0.00 | 0.00 | 0.00 |
| | WizardCoder | 0.00 | 0.00 | 0.00 | 0.00 | 0.00 | 0.00 | 0.00 | 0.01 |
| | Phi4 | 0.00 | 0.00 | 0.00 | 0.00 | 0.00 | 0.00 | 0.00 | 0.01 |
| | Gemma3 | 0.00 | 0.00 | 0.00 | 0.00 | 0.00 | 0.00 | 0.00 | 0.00 |

**Table 12: Performance Evaluation on Function Name Recovery with Different Architectures and Optimization Levels via Assembly Code.**

# F   LLM USAGE

We utilize LLM only for correcting and polishing the sentences.

