# OpenReview forum: "LLMs as Reverse Engineers? Not Yet on Types and Names"
_ICLR.cc/2026/Conference — ICLR 2026 Conference Withdrawn Submission_

### Official Review · Reviewer_TNfc · 2025-10-25

**Soundness:** 2
**Presentation:** 2
**Contribution:** 2
**Rating:** 2
**Confidence:** 4

**Summary:**

This paper systematically investigates the effectiveness of open-source mid-sized Large Language Models (LLMs) in reverse engineering tasks. While LLMs have shown promise in generating meaningful identifiers for functions, prior research has often focused narrowly on task-specific fine-tuning without justifying model choice. To address this, the authors evaluate CodeLlama, Llama 2, and DeepSeek-R1 across two key tasks—function name recovery and type inference. The results show that, without fine-tuning, these models achieve relatively low F1 scores, revealing their current limitations in handling binary analysis tasks. This study provides a clear benchmark for assessing the baseline capabilities of mid-sized LLMs in reverse engineering and underscores the need for targeted adaptation strategies to improve their performance in this domain.

**Strengths:**

Explore recent models.

**Weaknesses:**

What is the distinct contribution of this work compared to prior studies [1,2], which already provide more comprehensive experiments covering both function name recovery and summarization tasks?

The organization of the paper is also unclear, with many figures and tables placed in the appendix, which negatively impacts readability and overall presentation.

Moreover, the study focuses solely on general-purpose LLMs without examining the many existing fine-tuned LLM-based decompilers. Meanwhile, the findings offer limited insights and practical relevance for reverse engineering practitioners.

[1] Shang, Xiuwei, et al. "How Far Have We Gone in Binary Code Understanding Using Large Language Models." 2024 IEEE International Conference on Software Maintenance and Evolution (ICSME). IEEE, 2024.
[2] Shang, Xiuwei, et al. "An Empirical Study on the Effectiveness of Large Language Models for Binary Code Understanding." arXiv preprint arXiv:2504.21803 (2025).

**Questions:**

Include additional baselines, datasets, and evaluation tasks to provide a more comprehensive assessments.

---

### Official Review · Reviewer_dQcM · 2025-10-28

**Soundness:** 1
**Presentation:** 1
**Contribution:** 3
**Rating:** 2
**Confidence:** 4

**Summary:**

This work proposes a benchmark that evaluates (mid-sized) LLMs' capabilities on type inference and name recovery from stripped binaries (of C code). The evaluation pipeline is designed to automatically collect LLM's response into structured format (using an auxiliary LLM to format the collected response) and then run corresponding metrics for name recovery (semantic-based similarity) and type inference (exact match). Results show that none of the 9 mid-sized coding LLMs tested are useful for these two tasks. In total, 8 findings were reported, discussing the effects of fine-tuning, model size, training data, llm output formatting, etc.

**Strengths:**

S1: this work identifies mid-sized LLMs inability to solve the tasks of name recovery and type inference when given assembly and binaries.

S2: experiments were large-scale, tested on 9 different mid-size LLMs.

S3: this work provides some preliminary insights into how model size, fine-tuning, training data affect the performance of mid-sized LLMs on the two reverse engineering tasks.

**Weaknesses:**

W1: generally, everything is over-claimed. The author should make the claims specific to the target domain, as well as draw valid conclusions that can be supported by the evaluation results.

W1.1: overstated problem domain. This work only assessed mid-size LLMs on two tasks, however, both in the title and in various places in the writing it is mentioned that the reverse-engineering capability is tested. The authors should justify why and to what degree these two tasks are representative of all other reverse-engineering tasks.

W1.2: mid-sized LLMs are not representative of SOTA LLMs. As the evaluations are only done on mid-sized LLMs, **ALL** claims should be stated with respect to mid-sized LLMs, not extending to production level LLMs such as the ones used in coding agents. Although one of the experiment shows scaling from 7B to 34B of codellama does not bring significant improvement, it is not convincing enough if not even one of the SOTA models was tested, especially when all tested models are performing so poorly, and the scale on Figure 2 is tiny in absolute value so there does not seem to have any significant difference between the models with different sizes.

W2: motivation is not clear. The writing claims that there is a gap of under-justified choices of LLMs in the literature and that the proposed benchmark aims to improve this gap. However, it is not clear to the reader how this benchmark (which in an essence reveals LLMs' inability toward the two tasks) helps user choose an LLM for their coding task. In addition, it is not clear to the user why LLMs' ability in inferring types and variable names in assembly and binaries is helpful for end users. The author should motivate the readers a bit more on the significance of this evaluation.

W3: in the design of the automated pipeline, an auxiliary LLM is involved to clean the output of testing LLMs into structured responses. This design choice can be very problematic. First of all, it is not mentioned which LLM is used to do this task (which needs justification of the choice as well). Second, the auxiliary LLM might hallucinate, or not following instructions, so its output can be highly unstable or untrustworthy. The quality of this benchmark will then heavily depend on the capability of this auxiliary LLM, which is not desirable.

W4: poor writing qualities. Since these are minor, please refer to questions and suggestions for the details.

**Questions:**

Detailed questions and suggestions are as follows:

Q1: Line 73, any prior works that have tested llm type inference? Is there any reference on why types are measured by exact match? Any other metrics that can measure types (as the authors pointed out, the poor performance might be due to the exact match being too strict)?

Q2: Line 102-105, these related works are mentioned but not discussed at all. How are they related to the current work? What differentiates this work from them?

Q3: with a nearly zero result for all tested models, have the authors consider adjusting metrics (as mentioned before, maybe exact match is too strict) such that more insightful findings could be drawn to distinguish the capabilities?

#------------------#

Sugg1: all claims should be made on mid-sized llms instead of "current llms" as there is no evaluation on current llms (that are used by coding agents).

Sugg2: Line 114-121 seem to be very repetitive as these points were already discussed in the introduction. More detailed information/discussion should be included to illustrate the motivation.

Sugg3: Figure 3 and Figure 4 were mentioned in main text, however they are both in the appendix. Either move them into main text, or mention that they are in the appendix in the main text.

Sugg4: the citation of Jiang et. al. is missing complete information (publication year) Line 522.

Sugg5: line 260 - 267, the reader does not find a list of resulting figures/tables to be helpful for understanding the results and finding. Instead, each figure/table should be mentioned and referred in text within the respective discussions.

---

### Official Review · Reviewer_LjgB · 2025-10-31

**Soundness:** 2
**Presentation:** 1
**Contribution:** 2
**Rating:** 0
**Confidence:** 4

**Summary:**

This paper evaluates several open-source mid-sized LLMs on reverse engineering tasks such as function name recovery and type inference from stripped binaries. The authors present an automated benchmarking framework that standardizes heterogeneous model outputs via an auxiliary LLM for post-processing. Experiments compare nine models under various architectures and compiler optimization levels, before and after LoRA fine-tuning. Results show consistently low F1 scores (typically < 0.1), with fine-tuning offering inconsistent or marginal improvements. The authors conclude that current LLMs remain too limited for reverse engineering.

**Strengths:**

- The authors implement an automated pipeline that allows easy reproducibility and makes it easy to apply the setup to other models of interest.
- The dataset spans multiple architectures and compiler optimizations.
- The idea to provide a guideline on model selection for future experiments seems reasonable and useful. (But the paper does not sufficiently implement and show this.)

**Weaknesses:**

While the experimental procedure appears systematic, the study suffers from several methodological weaknesses.

Many larger models are excluded solely due to hardware or cost constraints, limiting the generalizability and diminishes the practical utility of the paper as a guide for model selection on reverse engineering tasks.

Important methodological details are missing from the main text. Most notably, a precise description of how F1 is computed for function name recovery and type inference, despite F1 being the central metric in most experiments. A clearer debate on the metric under investigation would make it easier interpreting the results better.

In the appendix, the authors state that evaluation depends to some extend on the random behaviour of the models. However, the evaluation metrics, particularly the F1 scores, are reported without details regarding confidence intervals. Furthermore, the evaluation relies solely on F1 scores, without considering a greater variety of metrics that could provide a more nuanced picture of model behaviour and error types. However, they raise awareness of that shortcoming in finding 2.

Moreover, several key design choices such as the prompting strategy, or the auxiliary LLM’s role in post-processing are insufficiently explained. For example, the paper mentions modifications to the prompt design without describing what was changed, which obscures the authors reasoning.

Another limitation lies in the narrow methodological scope. The authors only employ LoRA-based fine-tuning to improve model performance, without exploring alternative knowledge-editing techniques. Given the limited and inconsistent success of LoRA in their setup, the study would have benefited from comparing different methods.

Negative results are over-interpreted (See e.g. the debate around finding 4, or finding 6), e.g. leading the authors to draw disproportionately strong conclusions that current models cannot perform reverse-engineering tasks, rather than acknowledging possible shortcomings in their setup. Given the strictness of evaluation (e.g., exact type matches and sematic distance in naming) it is unclear whether poor performance reflects model incapability or inadequate task framing.

Furthermore, the findings come without offering actionable hypotheses for failure modes. (See e.g. the debate around finding 1, or finding 3.)
The manuscript is hard to follow and flows poorly; many sections read like disconnected bullet points rather than a cohesive narrative.
The related work section seems unbalanced. A few papers are discussed at some length, while many other relevant works are merely listed without sufficient depth.

While several external methods are mentioned, their descriptions remain mostly vague. Further studies such as Function-to-Style Guidance of LLMs for Code Translation (Zhang et al., 2025) could have provided further insights into how stylistic and structural cues enhance code understanding, which could be relevant to reverse engineering.

Tables 2–6 are not referenced at the points in the text where their results are discussed, and helpful visual aids (e.g., color coding to highlight results discussed by the authors) are absent. Figures are also consistently too small to read comfortably, which makes it harder to verify the arguments made around them.

Finally, from line 231 onwards citations stop abruptly mid-paper, and several referenced works are left without proper citation.

**Questions:**

- How was the auxiliary LLM used for normalization trained and validated? Could it bias the evaluation?
- Can the authors clarify how F1 scores were computed?
- What are the main sources of failure?
- Could you provide more qualitative examples or an error analysis to shed light on model limitations?

---

### Note · Authors · 2025-11-18

I have read and agree with the venue's withdrawal policy on behalf of myself and my co-authors.